# Multilevel Determinants of Integrated Service Delivery for Intimate Partner Violence and Mental Health in Humanitarian Settings

**DOI:** 10.3390/ijerph182312484

**Published:** 2021-11-26

**Authors:** M. Claire Greene, Clarisa Bencomo, Susan Rees, Peter Ventevogel, Samuel Likindikoki, Ashley Nemiro, Annie Bonz, Jessie K. K. Mbwambo, Wietse A. Tol, Terry M. McGovern

**Affiliations:** 1Heilbrunn Department of Population and Family Health, Columbia University Mailman School of Public Health, New York, NY 10032, USA; mg4069@cumc.columbia.edu (M.C.G.); cb433@cumc.columbia.edu (C.B.); 2School of Psychiatry, University of New South Wales, Sydney, NSW 2033, Australia; s.j.rees@unsw.edu.au; 3Public Health Section, United Nations High Commissioner for Refugees (UNHCR), Rue de Montbrillant 94, 1201 Geneva, Switzerland; ventevog@unhcr.org; 4Department of Psychiatry and Mental Health, Muhimbili University of Health and Allied Sciences, United Nations Road, Dar es Salaam P.O. Box 65001, Tanzania; likindikoki@gmail.com (S.L.); jmbwambo@gmail.com (J.K.K.M.); 5The MHPSS Collaborative, Rosenørns Allé 12, 1634 Copenhagen, Denmark; ane@redbarnet.dk; 6HIAS, Silver Spring, MD 20910, USA; annie.bonz@hias.org; 7Department of Public Health, Global Health Section, University of Copenhagen, Nørregade 10, 1165 Copenhagen, Denmark; wietse.tol@sund.ku.dk

**Keywords:** multisectoral, integrated care, intimate partner violence, gender-based violence, mental health, humanitarian

## Abstract

Inter-agency guidelines recommend that survivors of intimate partner violence in humanitarian settings receive multisectoral services consistent with a survivor-centered approach. Providing integrated services across sectors is challenging, and aspirations often fall short in practice. In this study, we explore factors that influence the implementation of a multisectoral, integrated intervention intended to reduce psychological distress and intimate partner violence in Nyarugusu Refugee Camp, Tanzania. We analyzed data from a desk review of donor, legal, and policy documents; a gender-based violence services mapping conducted through 15 interviews and 6 focus group discussions; and a qualitative process evaluation with 29 stakeholders involved in the implementation of the integrated psychosocial program. We identified the challenges of implementing a multisectoral, integrated intervention for refugee survivors of intimate partner violence at the structural, inter-institutional, intra-institutional, and in social and interpersonal levels. Key determinants of successful implementation included the legal context, financing, inter-agency coordination, engagement and ownership, and the ability to manage competing priorities. Implementing a multisectoral, integrated response for survivors of intimate partner violence is complex and influenced by interrelated factors from policy and financing to institutional and stakeholder engagement. Further investment in identifying strategies to overcome the existing challenges of implementing multisectoral approaches that align with global guidelines is needed to effectively address the burden of intimate partner violence in humanitarian settings.

## 1. Introduction

Intimate partner violence (IPV) is a significant threat to the health and wellbeing of women in humanitarian settings. Approximately one-third of ever-partnered women globally are estimated to experience physical and/or sexual IPV during their lifetime [1,2]. Research suggests a greater burden of IPV in humanitarian settings where lifetime prevalence estimates reach up to 73% among women [3,4]. IPV is a leading risk factor for poor health outcomes [5,6,7,8,9]. The complex, interrelated risk factors and consequences of IPV necessitate comprehensive solutions that are coordinated across sectors and across the prevention to response continuum.

Among the consequences of IPV are mental health problems, which are more prevalent in humanitarian settings, relative to non-humanitarian settings [10]. Epidemiologic data consistently identifies a relationship between IPV and mental health, including prospective studies which have found that IPV exposure is associated with an increased incidence of mental disorder [11,12,13,14,15,16,17]. Given the elevated burden of both IPV and mental health problems in humanitarian settings and their common co-occurrence [18], research has increasingly focused on identifying strategies to reduce the burden of these related public health challenges [19,20]. Psychological interventions for survivors of IPV have shown promising impacts on reducing the symptoms of common mental health problems [21]. Psychological services alone are unlikely to sufficiently address women’s violence protection needs [22]. This evidence has led to calls for combining psychological therapies with elements of advocacy, trauma-informed care, and linkages with other services that can more holistically address the needs of women affected by IPV [22]. However, research on multisectoral and integrated care models for IPV and mental health remains limited [20].

Multisectoral integrated care, which we define as ‘coordination and strategic collaboration across two or more sectors with the goal of achieving better health outcomes through collective action’ [23], can advance the health and wellbeing of individuals and populations in humanitarian settings. Leveraging economic and resource contributions across health, welfare, legal, and other services can address complex, inter-related public health problems, such as IPV and mental health, more efficiently while fostering collaborations that can advance humanitarian recovery and sustainable development [24]. This approach has been adopted in some high-income settings to provide services for IPV survivors [25,26], and is consistent with inter-agency gender-based violence guidelines for service delivery in humanitarian settings [27], as well as humanitarian mental health and psychosocial response guidelines [28]. Examples of integrated care delivery include one-stop centers that provide a range of health, psychosocial, legal, and other services, as well as integrating comprehensive services for GBV survivors within discreet entry points such as primary health facilities or general women’s protection programs [27]. For IPV survivors, integrated care can reduce the required time and visits to service providers, and more efficiently address the self-help and external support for needs that are inter-related. At the population level, integrated care models promote shared accountability for preventing gender-based violence in communities and systems [29].

While integrated models for delivering services and supports to women impacted by IPV and mental health problems are supported in existing guidelines [30], there are few examples of mental health interventions that have sought to prioritize an integrated, multisectoral approach in LMICs and humanitarian settings [20]. In this study, we aimed to examine what factors influence the implementation of a multisectoral, integrated intervention in a humanitarian context. To accomplish this objective, we described determinants, challenges, and opportunities for successfully implementing integrated IPV and mental health services using a case example from a feasibility trial of the *Nguvu intervention*, a psychosocial and protection intervention developed in Nyarugusu Refugee Camp, Tanzania. The Nguvu intervention was community-based and recognized the importance of ensuring that any psychological intervention for survivors of IPV was integrated within women’s protection services and in the context of a system providing other violence prevention and response programs. In this study, we distilled lessons regarding mental health and IPV services and how they worked together, including the challenges and barriers, to make recommendations for integrating services for IPV survivors in these settings.

## 2. Materials and Methods

### 2.1. Study Setting and Population

This study used data collected from Nyarugusu Refugee Camp, which is in north-western Tanzania. Shortly after its independence in 1961, Tanzania implemented an open-door refugee policy and has since hosted over 2.5 million refugees, primarily from the Democratic Republic of the Congo (DRC), Rwanda, Burundi, and Mozambique [31]. In the mid-1980s Tanzania shifted to establishing refugee camps with services provided primarily by humanitarian agencies and oversight by the Ministry of Home Affairs. This transition was formalized through the 1998 Refugees Act and the 2003 Tanzanian National Refugee Policy [32].

Nyarugusu, one of the oldest refugee camps in Africa, was established in 1996 in response to the 150,000 refugees fleeing the war in the eastern DRC to western Tanzania. Political violence in Burundi led to 123,285 refugees arriving in Tanzania by the end of 2015, with more arrivals in 2016 and 2017. All arriving Burundians were initially sent to Nyarugusu, creating overcrowding and short-term repurposing of schools and churches to shelters as its population reached 150,000 by the end of 2015 [33]. In 2016, The Tanzania Ministry of Home Affairs (MoHA), in collaboration with the United Nations High Commissioner for Refugees (UNHCR), re-opened two camps (Mtendeli and Nduta) that had been closed and, together with Nyarugusu, hosted 267,770 refugees [34]. By the end of 2017, the number of refugees and asylum seekers would grow to 359,000, with over 88 percent residing in camps, and Nyarugusu alone hosting 69,850 Burundians, close to 82,200 Congolese, and some 250 refugees from other nationalities [35].

The 2015 UNHCR Global Appeal for Tanzania noted “consistently high levels of SGBV [sexual and gender-based violence] and sexual exploitation and abuse in Nyarugusu camp, mainly resulting from harmful traditional practices affecting women and girls”, which it proposed to mitigate “through education, alternative energy solutions, women’s empowerment, and livelihoods” [36]. The monitoring of gender-based violence trends in Nyarugusu Refugee Camp drew on reports from survivors to implementing agencies that were compiled into a Gender-Based Violence Information Management System and supplemented with periodic assessments. Due to incomplete reporting, this monitoring system underestimates the true extent of gender-based violence in Nyarugusu and is used to monitor secular trends [37].

Available research from the eastern DRC and Nyarugusu has documented a high burden of gender-based violence, particularly IPV, and associated psychological sequelae in this population [37,38]. In addition to the common risk factors for IPV that are prevalent in Nyarugusu, there are unique aspects of the context that confer an elevated risk of IPV for refugee women. For example, qualitative research has revealed that the restrictions on livelihood activities and the inability of refugees to work is a source of family conflict and IPV, as fewer men have the opportunity to provide for their families and struggle with their changing roles and identity [37,39]. In our previous research in Nyarugusu, we found that mental health problems were common among survivors of IPV and resulted in substantial interpersonal and functional impairment [39,40]. Refugee women described deep sadness, stress, thinking too much, and fear as the most common problems affecting survivors of IPV in Nyarugusu [40]. These findings are consistent with population-based, epidemiologic studies conducted in the eastern DRC that have found a significantly higher prevalence of common mental health problems among women with a history of IPV, relative to women without a history of IPV [41].

Within this context, we developed and piloted a psychosocial intervention for female Congolese refugees in Tanzania between 2014–2017 that aimed to reduce psychological distress and IPV [40]. This project was initiated when Congolese refugees made up 94% of the population in Nyarugusu. Our project began before the influx of refugees from Burundi, and we did not have the resources to develop and adapt the intervention in multiple languages, cultures, and systems, and thus the intervention was focused on female Congolese refugees, most of whom had been living in Nyarugusu for the last two decades [42]. The Nguvu program integrated elements from an evidence-based psychological intervention, cognitive processing therapy [43,44,45], with a protection intervention, advocacy counseling, which is focused on increasing autonomy and empowerment, as well as strengthening linkages to community services [46,47]. Together, these intervention components provided the skills to manage the distressing thoughts that lead to emotional problems, empower women and promote autonomy, strengthen linkages to services across sectors, and facilitate safety planning and goal setting. These elements were combined into an eight-session intervention that was delivered by Congolese refugee women without prior specialized training in IPV and psychological interventions. Nguvu intervention facilitators were selected from among the refugee incentive workers affiliated with a humanitarian agency providing women’s protection services in Nyarugusu. The facilitators received an initial training from a clinical psychologist with expertise in trauma-informed care and a medical anthropologist with expertise in community psychiatric nursing and gender-based violence. After an initial intensive two-week training period, they received two refresher trainings delivered by mental health professionals and ongoing supervision by Tanzanian clinical psychologists [40].

### 2.2. Data Sources and Analysis

We combined an analysis of donor funding reports, legal and policy documents, a mapping of gender-based violence services in Nyarugusu, and program monitoring and evaluation data to explore the factors that influence the implementation of a multisectoral integrated intervention, such as Nguvu, in Nyarugusu. Focusing on the years 2015–2017, we reviewed UN funding and program reporting for Tanzania and the Burundi Regional Refugee Response; other donor and implementing agency self-reporting; Tanzanian laws and policy statements; and other grey literature that was published in English.

We also reviewed information from a mapping of organizations that were involved in IPV referral and support in 2016. The mapping was conducted by a medical anthropologist with the assistance of a Swahili interpreter and with support in recruitment and logistics from two national staff affiliated with an implementing agency providing GBV services in the camp. The mapping involved 15 individual interviews with a range of stakeholders across sectors (GBV officers, case manager, teachers, psychologists, protection officer, public health coordinator, police, *sungusungu*—local police, and medical providers), and six focus group discussions with members of the local Community Mediation Committee for Peace and Security (CMC, or *Amani na usalama*), gender-based violence volunteers, village and zone leaders, and other religious and cultural leaders. We used data from these qualitative interviews and focus groups as part of the study’s formative research to identify the institutions and stakeholders involved in IPV response, characterize the services they provided, and describe referral patterns among these entities. We continued to conduct focus groups until there was a consensus among the stakeholder groups about the GBV referral pathways and stakeholders.

To examine the factors that influenced implementation at the program level, we gathered data from a qualitative process evaluation which followed the completion of a cluster-randomized feasibility trial of Nguvu [48]. We conducted 29 semi-structured, in-depth interviews with stakeholders including 10 of the 158 Nguvu program participants, 10 intervention facilitators, 2 clinical and project supervisors, 3 representatives from implementing agencies, and 4 Congolese members of the community advisory board. We randomly selected five Nguvu program participants who were classified as ‘high attenders’ (i.e., attended more than six of eight sessions) and randomly selected five Nguvu program participants who were classified as ‘low attenders’ (i.e., attended fewer than four of eight sessions) to participate in the process evaluation. All other individuals who completed in-depth interviews were purposively selected by the research team based on their knowledge and role in the Nguvu program. All interviews were conducted in December 2017 in Swahili by trained ethnographic researchers. The semi-structured interview guide was designed to inquire about the relevance, acceptability, and feasibility of the Nguvu intervention using the parameters defined by the UK Medical Research Council’s process evaluation framework. Specifically, the framework examined contextual factors, the Nguvu implementation processes, and the mechanisms of impact [49]. Two independent researchers reviewed the interview transcripts and developed preliminary codes describing the determinants of the process evaluation outcomes. After achieving consensus on the final codebook, the researchers coded all study transcripts independently and resolved any discrepancies through discussion. Analysis of the data for this study involved selecting codes that referenced the multisectoral or integrated nature of the Nguvu intervention and mapping these codes onto a social ecological framework adapted to integrated and person-centered care [50] and Proctor’s implementation outcomes [51].

## 3. Results

Consistent with a social ecological analysis, we identified the determinants of successfully implementing multisectoral integrated IPV response services that were present within the structural context (*macro-system*), the inter-institutional context (*exo-system*), the intra-institutional context (*meso-system*), and the immediate social and interpersonal context (*micro-system*) [50]. In this paper, we describe these social ecological determinants in relation to the relevant implementation outcomes (Figure 1) [51].

### 3.1. Structural Context

In the context of IPV programming in a refugee camp setting, the structural determinants of multi-sectoral and integrated service delivery included the national and humanitarian governance and policy landscape, particularly as it relates to the legal context, access to justice, and financing of IPV services.

#### 3.1.1. Tanzanian Legal Context and Access to Justice

Tanzania acceded to the 1951 Convention Relating to the Status of Refugees and its Protocol Relating to the Status of Refugees and ratified the Convention on the Elimination of all forms of Discrimination against Women, all without reservations. It is also a signatory of regional refugee commitments, notably the 1969 Organization of African Unity (OAU) Convention Governing the Specific Aspects of Refugee Problems in Africa, which expanded the 1951 Refugee Convention’s protection triggers to include external aggression, occupation, foreign domination, and events seriously disturbing public order. It has yet to fully domesticate these treaties in national law and policy, which renders them unenforceable.

Tanzania’s plural legal system of customary, Islamic (in Zanzibar), and statutory law does not define or prohibit domestic violence and provides only limited protections against gender-based violence [52]. The 1998 Sexual Offences Special Provision Act revised the Penal Code provisions on rape, but did not extend them to criminalize marital rape unless the wife has a judicial separation order; while the Marriage Act of 2002, most recently revised in 2019, allows for the early and polygamous marriage of girls from age 15 (14 with court consent), and does not specify penalties for wife-beating. Neither the 1998 Refugee Act nor the 2003 National Refugee Policy have any provisions for gender.

The 1998 Refugee Act provides for detention and fines for refugees who leave their designated areas or work without a permit, severely restricting income-generating opportunities. The 2003 National Refugee Policy further limits work to small-scale activities within encampments. Refugee workers are permitted to work for modest incentives (i.e., refugee incentive workers), which are far lower than the salaries of the national staff [53].

In Nyarugusu Refugee Camp, a legal officer embedded within a protection implementing agency worked alongside the state prosecutor to support GBV survivors. Another non-governmental organization worked with survivors of gender-based violence on mediation and reconciliation. Additionally, the local Community Mediation Committee in Nyarugusu, *Amani na usalama* (‘peace and security’), was comprised of individuals appointed by cultural leaders to serve primarily as mediators for legal issues occurring within the community. Survivors of gender-based violence reported that they rarely seek legal assistance from the local mediation committee, which saw about three to four cases per month. This was attributed to the process being expensive, slow, and not culturally accepted by the family and community as a means to address IPV. The UNHCR provided training to the community on the formal legal process and applying a human rights-based approach.

#### 3.1.2. Financing

Humanitarian and development assistance to Tanzania are siloed by population and sector. Humanitarian aid for refugees is primarily organized through UN inter-agency funding appeals which are coordinated by the UNHCR operating under the UN’s Refugee Coordination Model, while multilateral and bilateral development aid for Tanzanian nationals is coordinated through the UN Resident Coordinator [54,55]. Through its annual Global Appeal, UNHCR requests for funding for the organization, to be allocated to the various refugee situations in the world. Additionally, UNHCR leads the development of Refugee Response Plans (RRP), which are comprehensive inter-agency plans for responding to refugee emergencies within a specific country, defining the financial requirements of all humanitarian actors. During refugee emergencies involving displacement to multiple countries, UNHCR coordinates the development of a Regional Refugee Response Plan (RRRP) with budgetary requirements by agency, implementing partner, and sector for each affected country. Significant funding shortfalls and plan revisions are common as conditions evolve, and many large donors provide UNHCR unrestricted funding or “softly earmarked” funding for a region or theme so that it can redirect monies as needed. Some donors and implementing organizations also provide funding outside of the response plan, which is not reflected in the UNHCR appeal expenditure tracking.

The Tanzania Country Appeal portion of the 2015 UNHCR Global Appeal listed gender-based violence prevention and response as one of two core areas it would prioritize in allocating funding, although its plan only measured the progress of response. It projected a cost of $1,446,109 to meet its target of providing appropriate support to 100 percent of known gender-based violence survivors; UNHCR’s total budget projection was $23,775,422 for roughly 60,000 Congolese and 190,000 Burundian refugees, later revised to $118.7 million to account for new arrivals [36]. The overall appeal for humanitarian assistance for Congolese and Burundian refugees in Tanzania was only 40% funded [33]. Actual UNHCR expenditures coded as gender-based violence prevention and support in 2015 was $738,809, about half of the requested funds for GBV. This figure includes 2015 expenditures under a series of Burundi RRRPs first issued in May 2015 and covering Burundian refugees in the Democratic Republic of Congo, Rwanda, and Tanzania. In Tanzania, the RRRP Protection sector strategy included the creation of “a multisectoral approach building on existing structures, response mechanisms and referral pathways [that] will ensure that SGBV prevention and response services will be provided”, including training of community gender-based violence focal points and sensitization on prevention and response. UNHCR budgeted $567,315 for gender-based violence prevention and response activities in 2016, but eventually spent $1,436,799 as the number of persons of concern increased 52%; that appeal was only 59% funded [34].

The difference in how violence against women is addressed for refugees versus nationals is apparent in the 2016–2021 UN Development Assistance Plan, known as UNDAPII [55]. For nationals, “Enhanced prevention of and response to violence against women and children” is a standalone outcome that combines implementation and monitoring of national plans of action, justice sector improvements, intersectoral case management and referral pathways, pilot and scale up of community-based norm-shifting approaches, and measures progress against multiple outcome and output indicators. For refugees, the plan for gender-based violence programming is more limited, and only addressed at the output level, with a focus on response and not prevention. Of the 20 output indicators for refugees and migrants, only one, “% of reported Sexual Gender Based Violence (SGBV) cases receiving psychosocial, medical, legal, material support in refugee camps”, addresses violence.

### 3.2. Inter-Institutional Context

Mapping the IPV services within Nyarugusu revealed a complex inter-institutional context comprised of both formal and informal services, which is similar to the organization of the IPV services described in other refugee camp settings [56]. We identified nine governmental or non-governmental organizations providing formal services to survivors of IPV in Nyarugusu Refugee Camp. These services included community-based rehabilitation (support centers, safe houses, psychosocial support), healthcare, legal assistance inside and outside of the camp (e.g., court and divorce cases), support with relocation and shelter, and security services provided by the police and camp management. In parallel, a local legal, cultural, health, and religious system also provided services to the survivors of IPV. This inter-institutional context influenced the implementation of the Nguvu intervention by complicating coordination and communication, stakeholder engagement and ownership, and the flexibility of resources to be applied across sectors to achieve shared objectives.

#### 3.2.1. Inter-Agency Coordination

The gender-based violence sub-working group within the protection sector coordinated the inter-agency and multisectoral services for IPV survivors in Nyarugusu. At the time of the study, they held weekly case conference and bi-weekly coordination meetings with agencies and entities who provided services relating to GBV. In addition to the GBV response programs, men’s discussion groups, community awareness raising sessions, and outreach were available as prevention programs during the implementation of Nguvu [57]. As noted elsewhere [40], the existence of these violence prevention programs was part of the rationale for the Nguvu intervention to focus on IPV survivors. In terms of the existing IPV response, coordinated care relied on referrals made by case managers. Barriers to coordinated service delivery included low rates of case detection, which was driven by community stigma and fear about the consequences of reporting incidents of gender-based violence, a lack of infrastructure to ensure confidentiality and coordinated care, limited access to justice services, and insufficient engagement with social protection and community systems [57].

Within the gender-based violence response plan, psychosocial support was primarily provided in the form of case management, which included basic counseling (i.e., active listening and emotional support). The Nguvu intervention aimed to fill a gap in services by introducing a more focused psychological support intervention into the IPV response and protection landscape. At the time of the study, the implementing agency had a mental health and psychosocial support (MHPSS) program in Nyarugusu, but they operated separately with limited referrals between mental health and IPV services. There were also efforts to train primary care providers to deliver basic mental healthcare in mhGAP around the time that Nguvu was being implemented [58], yet referral systems had yet to be formally established or adopted. In formative participatory research, as well as the subsequent process evaluation, stakeholders identified that an intervention which integrates components which focus on IPV and psychological distress was appropriate and complemented the existing services that were available in Nyarugusu. Having a functioning inter-agency gender-based violence working group with multisectoral objectives enabled the introduction of an integrated health and protection intervention (Nguvu) into the gender-based violence response. The MHPSS technical working group in Nyarugusu was intended to operate across sectors and be co-chaired by protection and health actors between whom there was a lot of bilateral engagement [59,60]. However, in practice, the MHPSS working group aligned most closely with the health sector and operated separately from the GBV working group, which led to coordination challenges. Even though it is common practice to have separate technical working groups in humanitarian settings, coordination challenges arise when there are no joint actions or standard operating procedures between working groups [61]. Also, multiple humanitarian coordination groups are challenging because of the additional meeting time required when human resources are limited.

#### 3.2.2. Inter-Institutional Engagement and Ownership

There was limited ownership of the Nguvu program by stakeholders who were not directly involved in the implementation, which compromised the adoption and sustainability of the program beyond the timeframe supported by the research. Nguvu participants, facilitators, research staff, and members of the community advisory board expressed the importance of continued delivery of Nguvu at the end of the project period. However, without coordinated stakeholder engagement and ownership, it was challenging to support ongoing implementation. When asked about integration and adoption, one Nguvu facilitator recommended that the intervention be independent, noting that:
“It should just be independent because there are so many organizations here and many projects are coming, and they are being integrated into [an ongoing program] so nothing is being done. If someone wants to improve their own thing, should you be dependent on being [adopted] by others? No, it is your duty to know the environment and how to implement the project.”—Facilitator.

The same Nguvu facilitator described that improving the adoption, implementation, and sustainability of a program like Nguvu, would require a stronger engagement of camp leaders and community members.

“We first start with camp leaders. We tell them that there is a project at the camp [that has been operating for] three years. The project came to treat the women and we see that the women have already been [helped]. What we ask now is for them to make the project [operate within the camp] … After we are done with the camp [leaders], we call women from the community. We need to hear from the women what Nguvu should do for them.”—Facilitator.

### 3.3. Intra-Institutional Context

The structured nature of the services delivered by implementing agencies (i.e., by specific donor-funded projects, with project-defined deliverables) complicated the integration of Nguvu into existing programming. Service delivery is often fixed, based on agency guidelines and intended project outputs that outline ways of working, including set response mechanisms and interventions. Adopting a multisectoral intervention into existing programming resulted in competing priorities for implementation staff. In addition, communication challenges existed with key decision makers, likely associated with a high staff turnover across levels (local, national, international) and confusion around ownership of the research. This catalyzed fluctuating engagement and support for the program at both the local and international level.

#### 3.3.1. Competing Priorities

Gender-based violence programming guidance states that mental health and psychosocial services are a critical and central component of any gender-based violence response, both short and long term [61]. Despite psychosocial care being central to a multisectoral response, it was challenging to integrate the Nguvu program within protection services due to resource constraints and reallocation of available resources owing to dynamic and competing priorities. Various stakeholders (implementing agency staff, facilitators, community members) recognized the relevance and value of a group-based integrated psychological intervention. However, accountability to activities and objectives that were often tied to existing donor-funded programs eclipsed the broad agreement that integrating mental health programming into women’s protection services advanced the overall goal of the gender-based violence response plan. This translated into difficulties securing the resources (e.g., private spaces) to implement the intervention and overburdening Nguvu facilitators with additional tasks.

The Nguvu facilitators were central to the successful implementation of the program. All were female incentivized refugee workers, a decision made to increase trust, limit language and other communication difficulties, and facilitate capacity building for a more sustainable IPV response within the refugee community. Intrinsic (e.g., desire to support their community) and extrinsic (e.g., positive reinforcement from Nguvu participants and community members for a valued new role in the community) sources of motivation promoted their continued commitment to the intervention, despite challenging structural and contextual barriers including legal limitations on refugee salaries and associated frequent renegotiation of contracts and benefits by facilitators.

The facilitators’ commitment to supporting violence-affected women, while operating within a system with limited ownership of the program, placed the facilitators in a difficult position. Many of the facilitators reported that, even though their role as an Nguvu facilitator was part of their incentive worker contract, these activities were often not prioritized or supported by their supervisors. This made it difficult for facilitators to lead Nguvu groups, which they were sometimes discouraged from doing due to competing priorities.
“The only challenge was from the [implementing agency] because the officers’ understanding [of Nguvu] differs. Sometimes you can ask for permission from the officer to go for an Nguvu session and they tell you to first do something else. Sometimes they refuse to let us go, or sometimes you might be on a session and they call you to tell you that you are needed at the office. You tell them ‘I am in the middle of an Nguvu session’, but they insist, so you have nothing to do. So I decide to deliver the session very fast and leave, but I promise the participants that for those who will find difficulties we can arrange another session tomorrow.”—Facilitator.

Some supervisors supported the integration of Nguvu into the services provided by the implementing agency and offered solutions to challenges that the facilitators encountered during implementation.
“There were challenges that happened, but they were solved by our administration. On the days that I had an Nguvu session, they gave me permission. So I was facilitating Nguvu sessions without any challenges on my side.”—Facilitator.

#### 3.3.2. Durable Intra-Institutional Engagement and Ownership

High rates of staff turnover reduced the predictability of support by institutional leadership, the key decision makers for the implementation of Nguvu. Variable levels of support translated into confusion about how facilitators should integrate Nguvu activities into their existing roles within the implementing agency and, ultimately, added to the facilitators’ workload.
“It is known that Nguvu is under [the implementing agency] so they have to allow us to fulfill the duties of Nguvu instead of giving us more duties of the [implementing agency] at the same time. Sometimes we experience challenges in fulfilling our responsibilities both at Nguvu and the [implementing agency].”—Facilitator.

One implementing agency staff recommended improved intra-institutional communication and engagement from the outset at all levels to promote the adoption of new programs to integrate within existing services and systems.
“When a program comes, it is better if it is introduced at the level of implementation. The top levels might work together and put together a nice plan, but when it comes to the field level, when it is time for its implementation, if we do not know its origin or its end and who it will benefit, how are we going to integrate it into our jobs? If we started together from the beginning and also plan together, this would have been much better”—Implementing agency staff.

This speaks to a possible disconnect between headquarter-based staff, who are often key decision makers, and country-based staff, who are responsible for the day-to-day operations of the program, including the implementation of research. This can be challenging in situations where large humanitarian agencies have many different levels of decision-making, and where staff turnover may further complicate communication between levels. Adopting multisectoral programming requires open communication at all levels and aids in implementation research not serving as a separate entity or add-on. Such communication challenges can also contribute to the above-cited disconnect between research and practice, and limit research uptake [19].

### 3.4. Social and Interpersonal Context

Nguvu participants described the integrated features of the intervention that benefited them. The survivors of IPV who participated in the program recognized the value of integrating services for mental health and IPV and described the integrated approach as being “beneficial to our homes and in our brains”. Nguvu was not seen as something that would be able to resolve IPV in Nyarugusu without a concomitant broader multisectoral and coordinated response: “This Nguvu project is ideal to decrease violence. Even if it will not [eliminate] it, it lowers it and one feels better.” Many stakeholders argued for integrating economic empowerment components, skills training and livelihood opportunities, and more intense engagement with men as strategies to prevent IPV. Several of these programs were already available as part of the gender-based violence response, yet results from the qualitative interviews suggest that many women were not aware of these programs.
“It is true that even men need it. Even though we have started the [men’s discussion] group, there is every reason to put more efforts into men in order to stop the violence. So we would like to look at a way that we could deal with the men so that the men can also know how violence can weaken themselves and their families.”—Implementing agency staff.

Nguvu participants described the need to provide education to men about the impact that IPV has on their families and communities, as well as the importance of changing norms and justificatory attitudes toward IPV. They mentioned the need for multiple modalities to effect this change, from individual interventions with men to community meetings. Financial dependence on men was also cited as a factor that made women more vulnerable to violence and led to women suggesting that income-generating or loan programs be integrated into IPV services. The power dynamics between women and their partners was a barrier to attendance for some women due to fear: “because of the focus on gender-based violence, they failed to ask for permission from their husbands”. To overcome this obstacle and prevent retaliation from partners who would resist women attending a program focused on IPV, some women described Nguvu to their partners as a program to educate them on women’s (mental) health. Having multiple related objectives within the Nguvu program enabled women to participate without revealing the focus on IPV response, which was a concern for some participants. Many other proximal socio-cultural and interpersonal factors were described in relation to the implementation of the Nguvu intervention, but not specifically its multisectoral, integrated features. Some key social, interpersonal, and cultural factors that influenced the successful implementation of Nguvu, which are described in detail elsewhere [62], included the cultural relevance and acceptability of the Nguvu intervention, group strategies and commitment to promote safety and confidentiality, and the motivation and supportive relationships among facilitators and Nguvu participants.

## 4. Discussion

Through a review of policy, financial, program documents, and qualitative research, we identified the determinants of multisectoral integrated IPV service delivery that existed at the structural, inter-institutional, intra-institutional, and immediate social and interpersonal levels. The key determinants of successful implementation of multi-sectoral, integrated interventions for IPV include access to legal assistance, financing, coordination, communication, engagement, and ownership. These determinants were found to influence the adoption, implementation, and sustainability of the Nguvu intervention in Nyarugusu Refugee Camp. The multilevel relationships among these determinants demonstrate the challenges and complexities of implementing and maintaining a multisectoral integrated program for IPV and mental health.

Many of the challenges experienced by the Nguvu program staff and facilitators may have been downstream consequences of the structural and coordination landscape. For example, program managers are often focused on reaching outcomes they see as central to the work on which they must report, which risks cross-cutting issues being given less importance. This challenge may have contributed to limited stakeholder engagement, ownership, and coordination in relation to the Nguvu program. Ultimately these consequences added to the strain and workload of humanitarian implementation staff and reduced the overall feasibility of this multisectoral integrated approach. Strengthening multisectoral systems and coordination mechanisms (e.g., multisectoral working groups) that facilitate integrated services for IPV survivors and the treatment of mental health as a cross-cutting issue is essential to fulfilling the principles and minimum standards outlined in inter-agency guidelines for both gender-based violence and mental health [27,28,59,60,63,64].

An important potential challenge that was not captured explicitly within the existing data was that Nguvu was a research activity and may have been perceived by implementers as an additional activity that usurped time and funds from the routine activities that they had planned. It is also possible that other factors that were not captured in this study may have contributed to these challenges. For example, issues related to the culture engrained within specific sectors (e.g., mandates, training backgrounds, language used, core shared values within sectors) may have reduced implementing partners’ willingness to allocate time and resources to services that were beyond the shared understanding of what activities a sector should engage in. Further efforts to engage actors across sectors in the design and development of programs that address their shared and respective interests may improve the adoption of multisectoral and integrated programs.

These study findings are comparable to previous research conducted on multisectoral approaches that has been designed to address a range of complex health conditions including chronic disease multi-morbidity [65,66], pandemic preparedness and response [67], HIV/AIDS [68], and nutrition [69,70]. These models have similarly identified challenges that undermine successful multisectoral integrated program planning. Some recognized challenges include distinct governance and accountability processes that make it difficult to work across sectors and organizations [71,72], separate financing mechanisms [73], misaligned incentives, ideological differences, weak partnerships and coordination [66], limited commitment and ownership of programs [74], and different organizational mandates and priorities [75]. This study is the first to examine the determinants of implementing multisectoral programming within a humanitarian setting, including for integrated mental health and protection services.

From a social justice perspective, it is critical that we tackle inter-related deprivations in key aspects of wellbeing (e.g., health, personal security, and equal respect), resulting from structured, interconnected patterns of disadvantage—such as is the case with violence against women—in coordinated, multi-sectoral ways [15]. Therefore, anticipating and developing strategies to overcome these challenges is essential to the effective implementation of multisectoral integrated models of care for IPV survivors that holistically and appropriately meet their needs. A recent review on the impact of psychological interventions on IPV in low- and middle-income countries identified three integrated interventions that aimed to reduce gender-based violence, reduce HIV and sexually transmitted infection (STI) risk, and improve mental health. Results suggested that the integrated interventions were not as consistently effective in reducing mental health problems as the dedicated mental health interventions [20]. While it is possible that integrating mental health and IPV interventions may have diluted or modified the core components of these interventions and, therefore, compromised their effectiveness, it is also possible that the complexity of implementing integrated interventions due to the challenges identified in this review may have also impacted these outcomes.

Further research on implementation strategies that are designed overcome these challenges is needed in order to introduce multisectoral integrated IPV service delivery models in humanitarian settings. Implementation strategies that align with existing guidelines and ensure adequate protection and support for women through integrated approaches are needed especially in contexts where there is not legal protection against IPV [27]. Rasanathan and colleagues (2017) developed a set of research questions to advance knowledge on the effective governance and administration of multisectoral programming for public health in low- and middle-income countries [24]. These priority research areas included examining how institutional structures and organizational culture facilitate opportunities for coordination and strategic collaboration, the role of incentives in motivating multisectoral collaboration, and how stakeholders interact to shape governance for health [24].

Research that embodies a multisectoral integrated approach to address IPV and mental disorder must also be contextualized for humanitarian settings. For example, as was described in our study findings, placing multisectoral interventions within the humanitarian coordination system and ensuring coordination across sectors was challenging. Furthermore, the dynamic nature of humanitarian settings, including the mobility of populations and agencies, resulted in instability in partnerships, as well as shifting priorities and resources to fit the needs of a rapidly changing situation (e.g., the influx of refugees from Burundi during project implementation). Future studies should embed operational research within implementing agencies to facilitate uptake and ownership, and to be responsive to the dynamic environment and evolving priorities within a complex humanitarian setting. The unique context of the humanitarian setting, including its governance, financing, and system-level organization and coordination mechanisms, requires targeted research to examine how to address the challenges and barriers that exist for multisectoral integrated programming. At the same time, it is important to draw on existing integrated service delivery and research models in the IPV and mental health fields outside these contexts [26]. A review of MHPSS research in humanitarian settings found that combining longer-term integrated programming with implementation and evaluation research informed by human rights, social justice, and feminist theory in emergencies can both meet priority humanitarian needs and determine how best to address the underlying social determinants of mental health, including gender-based violence [76].

In addition to providing recommendations for future research, our study complements prior findings regarding the conditions that can facilitate the successful implementation of multisectoral integrated services. First, policies and governance at the macro-system level must prioritize strategic collaboration models to enable stakeholders to work collaboratively to achieve shared goals (e.g., Collaborative Governance Framework) [71]. Second, co-financing and other funding models must recognize the value of integrated programming, and support agencies to engage in partnerships and collaborations across sectors. Longer-term and flexible funding that promotes strategic collaboration will facilitate agency level capabilities, such as the time and resources to invest in building and maintaining integrated services [73,77]. Designing program implementation plans using joint operational frameworks, integrated humanitarian response plans, and results-focused program outcomes can help to align incentives and promote accountability to broader multisectoral objectives [23,71,75]. Third, the development of multidisciplinary teams to work across organizational boundaries must be prioritized [78]. Finally, engaging communities and other stakeholders as central partners in planning for implementation can encourage program benchmarks and incentives to be aligned with population needs and culturally acceptable models of service delivery [71,79].

This research builds on a growing consensus that a multisectoral approach is needed to effectively address IPV, but these findings should be interpreted while considering the following limitations [27]. First, this study was conducted in a single refugee camp setting in Tanzania, and the findings may not be generalizable to other refugee camps or humanitarian settings. However, in this analysis, we aimed to specify the types of policies, programs, and contextual factors to enable the interpretation of these results in reference to other settings. Second, the data used for this study weren’t designed specifically to explore multilevel determinants of successfully implementing and maintaining multisectoral integrated IPV interventions. Therefore, the findings that we report may not comprehensively reflect the factors that contributed to the implementation of the Nguvu intervention or the perspectives of all relevant stakeholders. These qualitative findings could be strengthened through triangulation with quantitative analyses exploring these associations and the multilevel interactions that impact implementation of integrated services. Third, the financing and reporting systems often made it challenging to trace the funding and program outcomes specific to IPV and/or mental health. We only reviewed documents and reports that were published in English and may have missed legal, policy, or program reports published in other languages. Through triangulation of independent data sources in our analysis, we found substantial convergence and complementarity, thus increasing our confidence that this study identified salient and reliable themes related to multisectoral integrated IPV service delivery in refugee camp settings.

## 5. Conclusions

In this study, we examined factors that affected the successful implementation of a multisectoral, integrated service for mental health and IPV in Nyarugusu Refugee Camp. Findings from this study highlight the challenges and opportunities for implementing multisectoral integrated IPV programs within humanitarian settings. Despite consistent guidance recommending the use of multisectoral integrated service delivery models to address IPV, these approaches are often not implemented in practice within humanitarian contexts. While implementers may be inclined to apply these integrated approaches, we identified structural (e.g., policy and legal context, financing), inter-institutional (e.g., coordination, engagement), intra-institutional (e.g., competing priorities, ownership), and social and interpersonal (e.g., relevance, acceptability) barriers and challenges that may preclude their adoption. Future implementation research to design and test models of integrated service delivery, that is also embedded within humanitarian organizations, is needed to overcome these barriers and, ultimately, improve the alignment of IPV service delivery to prevent IPV, enhance safety, and promote optimal mental health and functioning for women.

## Figures and Tables

**Figure 1 ijerph-18-12484-f001:**
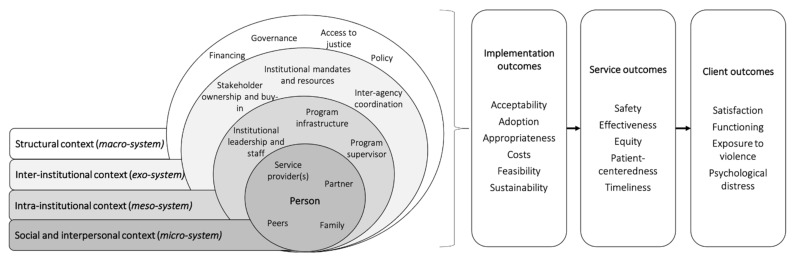
Multilevel determinants of integrated service delivery for IPV and mental health in Nyarugusu Refugee Camp (adapted from Proctor et al., 2011; Woolcott et al., 2019) [50,51].

## Data Availability

The data underlying the results will be made available upon reasonable request to the first author (mg4069@cumc.columbia.edu).

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
