# Peer review of "Multilevel Determinants of Integrated Service Delivery for Intimate Partner Violence and Mental Health in Humanitarian Settings"

_ijerph, 2021, doi:10.3390/ijerph182312484_

Round 1

Reviewer 1 Report

This well written manuscript explores factors that influence the implementation of an intervention to reduce psychological distress and intimate partner violence in a refugee camp in Tanzania. The questions are of important scientific and public health relevance, however the conclusions are somewhat straightforward and expected, although important. Some specific areas to review include:

  • Methods:
    1. Additional information on the language of some of the primary data sources, especially as it relates to the language of the analyses would be helpful (were donor funding reports, legal, and policy and mapping services done in English). FGD in English / Swahili?
    2. Additional information on the FGD would be helpful, were these designed just for this study or were data from FGDs repurposed for this study? How was saturation confirmed and how were groups formed? Areas where the implementation strayed away from FGD best practices should be flagged in the limitations paragraph in the Discussion.
    3. Additional information about the type of GBV services would be helpful, who were their clients, funding, longevity, approach, etc.
  • Results:
    1. Figure 1 is a very nice figure. Following the figure in the results section – i.e. having Financing, Governance, Access to Justice, and Policy as subheadings – or at least detailed would improve the manuscript. As written, it isn’t clear why some are selected for further elaboration as opposed to others (e.g. only legal (as proxy for access to justice and governance?) and financing are detailed in structural level).
  • Discussion:
    1. Lines 508-517; this paragraph could be strengthened by highlighting what this piece contributes beyond previously published work.
    2. Greater attention should be paid in the limitations to this being work from one location and discussion of implications across contexts is warranted
  • Minor typographical errors
    1. Introduction: line 49 missing hyphen or comma around “humanitarian relative to non-humanitarian settings”

Reviewer 2 Report

Kindly revisit the title. The title is too long and this may likely lose focus of the study confuse any reader.

Reviewer 3 Report

Dear Author(s),

Thank you for submitting this manuscript, which addresses a very relevant and timely topic.

ABSTRACT: I would not separate the abstract into different sections.

INTRODUCTION: Clearly state one or more research question(s), better highlighting the originality of your work with respect to other academic papers on the topic. Include here some brief notes on the methodology adopted and on the structure of the following parts.

MATERIALS AND METHODS – A literature review section is missing. Please include a paragraph with recent literature on the topic.

METHODOLOGY, RESULTS AND DISCUSSION: In my opinion, these sections are well addressed.

CONCLUSIONS: Clearly answer to the research question(s) raised in the introduction and then include considerations on the possible generalizability of the results, as well as on the practical implications of your work.

EDITORIAL GUIDELINES: Please, adapt the text according to MDPI editorial guidelines, mainly with respect to references.

Best wishes

Round 2

Reviewer 3 Report

The manuscript has consistently improved with respect to the previous version and can be now accepted in its current form. Congratulations!